# Metabolomic Profile of Açai (*Euterpe oleracea* Mart., *Euterpe precatoria* Mart.), Mirití (*Mauritia flexuosa* L.), and Cupuassu (*Theobroma grandiflorum* (Wild. ex Spreng.) Schum) from Colombian Amazon: Insights into Nutritional Composition and Ripening Dynamics

**DOI:** 10.3390/ijms27010410

**Published:** 2025-12-30

**Authors:** Manuel Salvador Rodríguez, Aida Juliana Martínez León, Lina Sabrina Porras, Iván Alejandro Giraldo, Esmeralda Rojas, Fredy Eduardo Lavao, Kaoma Martínez

**Affiliations:** 1Departamento de Ingenieria Civil y Ambiental, Universidad de los Andes, Bogotá 111711, Colombia; ai-marti@uniandes.edu.co (A.J.M.L.); lporras@uniandes.edu.co (L.S.P.); ia.giraldo90@uniandes.edu.co (I.A.G.); 2Gobernación del Vaupés, Planeación Departamental, Calle 15 #14–18 Barrio Centro A Mitú Vaupés, Mitú 970001, Colombia; planeaciondepartamental@vaupes.gov.co (E.R.); cienciatecnologiainnovacion@vaupes.gov.co (K.M.); 3Gobernación del Vaupés, Secretaria de Agricultura y Desarrollo Productivo, Carrera 10 #9–32, Mitú 970001, Colombia; secretariadeagricultura@vaupes.gov.co

**Keywords:** carbohydrates, flavonoids, alcohols and polyols, carboxylic acids, secondary metabolites

## Abstract

Amazonian fruits are increasingly recognized for their functional properties due to their rich composition of bioactive metabolites. While species such as *Euterpe oleracea* Mart., *Euterpe precatoria* Mart., *Mauritia flexuosa* L., and *Theobroma grandiflorum* (Wild. ex Spreng.) have been extensively studied in countries like Brazil, research on these fruits in Colombia remains limited. This study aimed to characterize the secondary metabolites in freeze-dried pulp and seed samples of açaí, mirití, and cupuassu at different ripening stages, collected in Mitú (Vaupés, Colombia). Eleven samples of different fruits were collected and analyzed by untargeted metabolomics. Untargeted metabolomic profile was performed using LC-QTOF-MS and GC-QTOF-MS techniques. The pulp of *M. flexuosa* showed high concentrations of galacturonic acid. In *T. grandiflorum*, both pulp and seeds contained significant levels of galactose and citric acid. Notably, flavonoids such as fisetin and kaempferol-O-rutinoside were abundant in the pulp of *E. oleracea* and *E. precatoria*. Additionally, quinic acid and mannitol were detected in the unripe stages of *Euterpe* and *Mauritia* species. The diverse profile of secondary metabolites identified in these Colombian Amazonian fruits underscores their functional and nutritional potential. These findings support further exploration of native species for food, nutraceutical, and industrial applications, contributing to the valorization of Amazonian biodiversity.

## 1. Introduction

The Amazon region contains the largest continuous expanse of forest on earth and harbors nearly 30% of the planet’s biodiversity. It is home to a wide variety of flora and fauna, including numerous species unknown to science. Among these, Amazonian fruits represent a significant component of the region’s biological richness [1]. The Amazon hosts approximately 35 genera and over 170 species of palm trees, of which at least 96 produce edible fruits. Notable species include açai (*Euterpe oleracea* Mart. (Martius, 1824) and *Euterpe precatoria* Mart. (Martius, 1842)) and miriti (*Mauritia flexuosa* L., (Linnaeus filius, 1782)) [2]. Additionally, cupuassu (*Theobroma grandiflorum*, Wild. ex Spreng. (Schum, 1886)), a close relative of cacao (*Theobroma* sp.), has gained relevance in agroforestry systems and contributes nutritional value to local diets. These fruits have attracted scientific interest due to their bioactive compounds with antioxidant, immunomodulatory, and anti-inflammatory properties [3]. However, most studies focus on Brazilian populations, overlooking the ecological variability of the broader Amazon. Environmental heterogeneity, species distribution, and complex interactions among edaphic, climatic, and anthropogenic factors can influence fruit phytochemistry across regions [4]. The department of Vaupés, located in southeastern Colombia within the Amazon basin, exhibits high biological diversity, with tropical forests containing over 120 plant species per hectare. Palms are a key element of local ethnobotany, used in housing, utensils, food, and cosmetics. Dominant species include *E. precatoria*, *E. oleracea*, *M. flexuosa*, *Oenocarpus bataua*, *Bactris gasipaes*, *Attalea maripa*, and *Astrocaryum chambira*, as well as fruit-bearing shrubs such as *Averrhoa carambola* and *T. grandiflorum* [5]. *E. oleracea* and *E. precatoria* are helophytic palms inhabiting humid forest from sea level to 2000 m. Their natural range extends from Belize to Brazil and Bolivia. These species grow up to 25 m in height with trunks measuring 23 cm in diameter [6]. Their fruits, known as açai, are spherical, purple, and approximately 1.1 cm in diameter. They exhibit high levels of anthocyanins, carotenoids, and dietary fiber, and are recognized for their notable antioxidant capacity. The fruits have moderate fat content and are low in carbohydrates and moisture [7]. *M. flexuosa*, from the *Arecaceae* family, is widely distributed in Bolivia, Brazil, Colombia, Ecuador, the Guianas, Peru, Trinidad and Tobago, and Venezuela. It can reach heights of up to 40 m, with trunk diameters ranging from 30 to 60 cm. Its globose fruit, called miriti, measures 4–6 cm in length and 3–5 cm in diameter, typically containing a single seed. The pulp is rich in β-carotene, lipids, fiber, and vitamin C, and serves as a significant source of provitamin A and energy. It is also notable for its high iron, copper, and potassium content. Lipids are predominantly composed of oleic acid (omega-9) [8]. *T. grandiflorum*, of the *Malvaceae* family, is widely distributed across non-flooded, well-drained areas of the Amazon. Its fruit, cupuassu, is ellipsoid (12–25 cm in length), with a hard, woody exocarp. The white pulp is rich in phosphorus and pectin, and contains moderate levels of calcium and vitamin C. Its lipid profile, dominated by oleic and stearic acids, underpins its use in natural moisturizing and softening agents [9]. Over the last decade, secondary metabolites from Amazonian fruits have received growing attention for their potential in food, pharmaceutical, and cosmetic applications [2]. In addition to the well-characterized phytochemical composition of açai, miriti, and cupuassu pulps, previous research has examined the chemical and bioactive profiles of the seeds of these fruits. Açai seeds have been reported to contain abundant dietary fibers and phenolic antioxidants such as catechin, epicatechin, with fatty acids including oleic and linoleic acids [10]. *M. flexuosa* seeds have been reported to contain abundant crude fiber, polyphenols, and flavonoids. For *T. grandiflorum* seeds, existing research indicates appreciable amounts of lipids, fiber, and phenolic compounds.

Despite substantial research in Brazil, data remain scarce for the same species in other Amazonian countries, including Bolivia, Colombia, Ecuador, Guyana, French Guiana, Peru, Suriname, and Venezuela. Accordingly, this study aims to characterize the bioactive compounds of three Amazonian fruits—açai, miriti, and cupuassu—at various stages of ripeness, collected in Mitú, Vaupés (Colombia), using an untargeted metabolomic approach via LC-QTOF-MS and GC-QTOF-MS. Fruit ripening is a developmental process that influences the biosynthesis, degradation, and distribution for bioactive compounds, as well as attributes like color, aroma, and flavor [11]. In fruits such as acai, mirití, and cupuassu, ripening-associated metabolic changes are linked to variations in phenolic compounds, anthocyanins, carotenoids, lipids, and organic acids. These changes can determine marketability and consumer preferences, as fruits harvested at optimal ripeness exhibit desirable pigmentation and nutritional value while under- or over-ripe fruits may present reduced shelf life [12]. However, assessing bioactive compounds across ripening stages remains challenging due to asynchronous ripening, sensitivity of secondary metabolites to environmental conditions, and post-harvest handling and processing [13,14].

## 2. Results

### 2.1. Metabolomic Profile of Analyzed Fruits

The untargeted metabolomic profile of *M. flexuosa* (miriti pulp) identified 110 distinct metabolites in the ripe stage and 96 in the unripe stage (Appendix A). The predominant classes in both maturity stages were carbohydrates and alcohols, followed by lower abundances of fatty acids, hydroxy acids, and amino acids. Carbohydrates accounted for 64.4% in ripe and 53.5% in unripe samples. Alcohols and polyols comprised 13.0% and 21.3%, respectively (Figure 1).

For *E. oleracea* (açai pulp), 100, 103, and 105 compounds were identified in ripe, intermediate, and unripe stages, respectively. In *E. precatoria*, 99 compounds were detected in ripe and 101 in unripe samples (Appendix A). Across both species, flavonoids were the most abundant class, followed by alcohols and polyols, fatty acids, carbohydrates, and phenolic acids (Figure 1). In *E. oleracea*, flavonoid content reached 69.9%, 50.2%, and 22.9% in ripe, intermediate, and unripe stages, respectively. In *E. precatoria*, flavonoids were present at 60.6% and 33.6% in ripe and unripe samples. While compound distribution varied between species, flavonoids consistently dominated, with increasing concentrations during fruit ripening.

In *T. grandiflorum* (cupuassu), metabolomic analysis of pulp identified 43 compounds in ripe and 61 in unripe samples. For seeds, 77 and 70 compounds were detected in ripe and unripe states, respectively (Appendix A). In pulp, carboxylic acids were the dominant class (40.5% ripe; 31.2% unripe), followed by carbohydrates (36.5% and 31.1%), hydroxy acids, and alcohols. Minor components (<5%) included amino acids, flavonoids, fatty acids, and amines.

In cupuassu seeds, carbohydrates, carboxylic acids, and flavonoids were predominant. Carbohydrates accounted for 48.4% and 45.6% in ripe and unripe seeds, respectively; carboxylic acids reached 15.6% and 16.9%, while flavonoids were detected at 12.0% and 14.6%. Other detected classes—alcohols and polyols, amino acids, fatty acids, and amines—remained below 8% in both maturation stages (Figure 1) (Appendix A).

### 2.2. Bioactive Compounds

The principal bioactive compound classes identified across the analyzed fruits were carbohydrates, flavonoids, alcohols and polyols, and carboxylic acids. Carbohydrates were most abundant in *M. flexuosa* samples, followed by *T. grandiflorum* seeds and pulp. In contrast, *E. oleracea* and *E. precatoria* exhibited lower carbohydrate concentrations (Figure 2).

The predominant carbohydrate was galacturonic acid, representing 28.2% and 22.3% in ripe and unripe *M. flexuosa* samples, respectively. Fructose was also prominent in unripe *M. flexuosa* (21.6%) and to a lesser extent in ripe samples (7.8%). Conversely, allose showed higher abundance in ripe samples (13.6%) compared to unripe (0.7%).

In *T. grandiflorum* seeds, galactose was the major carbohydrate, accounting for approximately 18.0% in both stages of maturity. In pulp, galactose was present at 12.2% and 11.9% in unripe and ripe states, respectively. Glucuronic acid was also notable in seed samples, constituting roughly 16.0% across both stages. Additionally, gluconic acid was detected in seeds, with levels of 12.6% (ripe) and 10.9% (unripe). Lactose was predominantly found in ripe seeds (6.3%). Minor compounds such as gentiobiose and glycerol were detected at low levels, while fructose, saccharic acid, and lactulose were absent. Flavonoid content was highest in the ripe stages of *E. oleracea* and *E. precatoria*, followed by the intermediate stage of *E. oleracea*, and subsequently the unripe stages of both species. In *T. grandiflorum*, flavonoid concentrations were greater in seeds compared to pulp. In contrast, flavonoid levels in *M. flexuosa* pulp were negligible (Figure 3).

Among the dominant flavonoids in açai, fisetin was the most abundant compound in ripe *E. oleracea* (23.4%), followed by ripe *E. precatoria* (16.5%), and intermediate *E. oleracea* (11.9%). Kaempferol-O-rutinoside was detected at 22.8% in ripe *E. precatoria*, 19.3% in ripe *E. oleracea*, and 8.2% in intermediate *E. oleracea*. Cyanidin-O-galactoside reached 12.5% in ripe *E. oleracea*, 11.0% in intermediate *E. precatoria*, and 10.9% in ripe *E. precatoria*. Catechin was present at 7.5% in unripe *E. precatoria* and 4.0% in its ripe stage (Figure 3).

As shown in Figure 4, alcohols and polyols were most abundant in the unripe stages of *E. oleracea* and *E. precatoria,* followed by unripe *M. flexuosa*. Lower concentrations were detected in pulp and seed samples of *T. grandiflorum*. The major compound across all species was quinic acid, followed by mannitol, myo-inositol, sorbitol, shikimic acid, and hydroxyethylcatechol (Figure 4). Quinic acid was particularly prevalent in unripe *E. precatoria* (23.2%) and *E. oleracea* (13.6%). In ripe fruits, concentrations decreased to 10.6% in *E. precatoria* and 1.8% in *E. oleracea*. Mannitol was consistently present in all maturity stages of *Euterpe* species and in the unripe *M. flexuosa*, with the highest level observed in unripe *E. oleracea* (16.6%), followed by *M. flexuosa* (12.0%). Myo-inositol was primarily detected in *M. flexuosa* and *T. grandiflorum*, reaching 7.1% in ripe *T. grandiflorum* pulp and 7.0% in ripe *M. flexuosa*. Unripe samples of both species showed lower values (4.1% in *M. flexuosa* and 3.8% in *T. grandiflorum*). Shikimic acid was most abundant in *M. flexuosa*, with 4.5% and 3.8% in ripe and unripe stages, respectively. Hydroxyethylcatechol was present at levels below 1%.

Carboxylic acids were the most abundant compound class in *T. grandiflorum*, followed by *M. flexuosa*, *E. oleracea*, and *E. precatoria* (Figure 5). Citric acid was the predominant organic acid in *T. grandiflorum* pulp, representing 35.4% in ripe and 24.5% in unripe samples. In seeds, citric acid concentrations were lower—13.6% in ripe and 12.2% in unripe samples. In *M. flexuosa*, citric acid was present at 1.5% and 1.1% in ripe and unripe pulp, respectively.

Glutaconic acid was also detected in *T. grandiflorum* pulp, with levels of 3.3% (unripe) and 2.7% (ripe). In seeds, concentrations were 2.1% (unripe) and 0.6% (ripe). Other carboxylic acids such as isonicotinic acid, maleic acid, and hydroxyphenyl acid were found at levels below 1% across all fruit samples. Additional bioactive compounds—including amines, phenolic acids, fatty acids, and amino acids—were detected at comparatively low concentrations in all analyzed fruits. A comprehensive summary of these metabolites is provided in Appendix A.

## 3. Discussion

Amazonian fruits are rich in micronutrients and bioactive compounds, including antioxidants and phenolics. Several studies have reported health-promoting properties associated with these fruits, including anticancer, anti-inflammatory, anti-obesity, antihypertensive, and antidiabetic effects [15,16].

In the present study, the carbohydrate content was highest in the ripe pulp of *M. flexuosa* and *T. grandiflorum*, with *T. grandiflorum* seeds presenting great value. These values align with those reported by [8], who found carbohydrate concentrations ranging from 69.7 to 71.0% DW in *M. flexuosa* pulp and 35.6 to 45.2% DW in lyophilized samples. For *T. grandiflorum*, ref. [3] reported concentrations of sugars ranging from 11 to 49 g 100 g^−1^ in pulp and 13.6 to 26.4 g 100 g^−1^ DW in seeds. Variability may be attributed to maturity stage, environmental conditions, and genotypic differences. Notably, both cited studies were conducted in the Brazilian Amazon.

Galacturonic acid was the major carbohydrate in *M. flexuosa*. Higher levels (47.4–57.6% FW) were previously reported in ripe pulp by [17]. Additional compounds identified in *M. flexuosa* included glucose (4.20 g 100 g^−1^), maltose (37.33 g 100 g^−1^), and xylose (4.91 g 100 g^−1^). Refs [18,19] also reported rhamnose (11.3–20.5 mol% DW), fructose (0.8 mol% DW), arabinose (163.1–380.6 mol% DW), xylose (9.0–156.6 mol% DW), mannose (2.8–8.2 mol% DW), galactose (21.9–23.2 mol% DW), glucose (3.0–117.7 mol% DW), and uronic acids (36–104.8% *w*/*w* DW), which are consistent with findings in this study. Galacturonic acid is not only an energy source but also contributes to gut health and metabolic regulation. It is widely used as a gelling agent and is a key component of pectin, a soluble dietary fiber known to modulate glycemic response, improve lipid profiles, and support gut microbiota balance. Its presence in high concentrations in *M. flexuosa* suggests potential prebiotic effects and applications in functional food formulations [18,19].

Galactose was predominant in *T. grandiflorum* seeds and present at considerable concentrations in the pulp. These levels are comparable to those reported by [20], who observed 13% DW galactose in cupuassu pulp. Other carbohydrates detected in *T. grandiflorum* pulp included sucrose (108 mg g^−1^), glucose (24 mg g^−1^), and fructose (36 mg g^−1^), aligning with the carbohydrate profile observed in this study.

On the other hand, flavonoids were detected in all fruits, with the highest levels found in ripe and intermediate pulp of *E. oleracea* and *E. precatoria*. Fisetin, kaempferol-O-rutinoside, and cyanidin-O-galactoside were the predominant flavonoids in these species. Fisetin, a flavonol, exhibits chemopreventive and chemotherapeutic activity against various cancers. Kaempferol, a potent antioxidant, surpasses vitamins C and E in activity and is associated with benefits in cancer, diabetes, obesity, hepatic disorders, and colitis. Kaempferol-O-rutinoside also displays antioxidant and antidiabetic properties by enhancing glucose uptake and insulin release [21].

Kodikara et al. [22] highlighted that fisetin, kaempferol, and luteolin share the same parent ion (*m z*^−1^ 285.0404), but differ in fragment ion patterns due to structural and stability differences. These similarities pose challenges in unambiguous compound identification despite advances in mass spectrometry. This could explain why fisetin was uniquely identified by [23], while kaempferol and its glycosylated derivatives are consistently reported in açai [23,24,25].

Studies by [7,12] on Brazilian Amazonian fruits reported cyanidin-3-glucoside (4.94 mg 100 g^−1^), cyanidin-3-rutinoside (17.9 mg 100 g^−1^), and isovitexin (2.66 ± 0.14 mg 100 g^−1^). In Colombia, ref. [26] identified isovitexin at 12.0 ± 4.8 mg 100 g^−1^ in açai pulp. These findings are consistent with this study, where cyanidin-O-galactoside was found in *E. oleracea*, *E. precatoria*, and *T. grandiflorum*, with concentrations ranging from 12.5% to 0.2%. Isovitexin was detected only in *E. oleracea* and *E. precatoria*. Both compounds are potent antioxidants that protect against oxidative damage [24].

Alcohols and polyols were identified in all samples, with highest concentrations in unripe *E. oleracea*, *E. precatoria*, and *M. flexuosa*. Quinic acid, mannitol, and myo-inositol were the dominant compounds in these species. Previous studies reported quinic acid in *M. flexuosa* pulp (230.74 mg g^−1^) and caffeoylquinic acid in *Euterpe* leaf and root extracts. The detection of quinic acid, mannitol, and myo-inositol in unripe stages of *Euterpe* and *Mauritia* species expands the nutritional profile of these fruits. Quinic acid has been associated with anti-inflammatory and neuroprotective effects, while myo-inositol plays a role in insulin signaling. Mannitol, a sugar alcohol, may contribute to osmotic balance and has potential as a low-glycemic sweetener [27,28,29,30].

Carboxylic acids were present in all species analyzed, with highest concentrations in *T. grandiflorum*. Citric acid dominated in pulp and was also found in seeds [3,31]; studies previously reported 176 mg g^−1^ of citric acid in *T. grandiflorum* pulp and lower levels in seeds (1.61 mg g^−1^). Citric acid is widely used as a food additive due to its acidifying and preservative properties [32].

The metabolites identified in this study play important roles in the physiological and biochemical changes that occur during fruit ripening. Carbohydrates, such as galacturonic acid, a major component of pectin, contribute to cell wall softening. Flavonoids, including fisetin, kaempferol-O-rutinoside, are linked to pigmentation and antioxidant defense. Carboxylic acids, like citric acid, influence flavor by modulating acidity [2,11].

The metabolomic profile of these Colombian Amazonian fruits reveals a matrix of compounds with synergistic effects on human health, linking specific metabolites to nutritional endpoints such as glycemic control, antioxidant status, and anti-inflammatory activity.

## 4. Materials and Methods

### 4.1. Samples

A total of eleven samples of açai (*E. oleracea* and *E. precatoria*), miriti (*M. flexuosa*), and cupuassu (*T. grandiflorum*) fruits were collected over a three-month period in Mitú, Vaupés, Colombia (Latitude: 1°14′54.36″ N, Longitude: 70°14′24.66″ W). The study was approved by the “Collection Framework granted to Universidad de los Andes by Resolution No. 002377, 2024-RCM0014-00-2024 and Addendum No. 2 of the Framework Contract for Access to Genetic Resources and Derived Products No. 288, 2020, file RGE338-2.” Post-harvest, the fruits were transported to the laboratory in plastic bags. Manual selection was performed based on morphological characteristics and ripening stage, as detailed in Table 1.

Açai fruits from both species were processed at distinct ripening stages. For *E. oleracea*, three maturation stages—ripe, intermediate, and unripe—were assessed, whereas for *E. precatoria*, only ripe and unripe stages were included. Selected fruits were washed and disinfected, then immersed in water at an ambient temperature for 12 h to facilitate pulp detachment. Pulp extraction was carried out using a food-grade mechanical pulping machine consisting of a stainless-steel perforated drum (mesh size: 1.0 mm), powered by a motor of 4 HP. The system enabled efficient separation of pulp from seeds and fibrous residues. The resulting pulp was transferred into Falcon tubes and stored at −80 °C for 24 h prior to freeze-drying. Miriti (*M. flexuosa*) fruits were analyzed at two ripening stages (ripe and unripe). Following washing and disinfection, fruits were immersed in water at 35 °C for 12 h to assist epicarp removal. Pulp was separated from seeds and epicarp using a pulping machine. The pulp was stored in Falcon tubes at −80 °C for 24 h before freeze-drying [33,34].

For *T. grandiflorum*, both pulp and seed samples were collected at ripe and unripe stages. Pulp was manually extracted by separating it from the woody exocarp. Subsequently, it was stored in plastic bags at −80 °C for 24 h and subjected to freeze-drying. Seeds were dried using a food-grade desiccator and then milled to obtain a fine powder, which was stored in Falcon tubes until analysis.

### 4.2. Metabolomic Profile by LC-QTOF-MS

Freeze-dried samples were used for metabolomic characterization. A total of 10 mg of each sample was mixed with 400 µL of MeOH/water (1:3). The samples were mixed by vortex at 3200 rpm for 15 min, followed by 10 min in ultrasound. Finally, the samples were centrifuged at 1400 rpm for 10 min. The supernatant was filtered with PTFE membranes with 0.22 µm diameter. The Agilent Technologies 1260^®^ Agilent Technologies Inc., Santa Clara, CA, USA, Liquid Chromatography system coupled with a Q-TOF 6545 time of flight quadrupole mass analyzer with electrospray ionization (ESI) was used A total of 1 µL aliquot of each sample was injected into a C18 column (InfinityLab Poroshell 120 EC-C18 (100 mm × 3.0 mm, 2.7 µm) Agilent Technologies Inc., Santa Clara, CA, USA) at 30 °C with a gradient elution composed of 0.1% (*v v*^−1^) of formic acid in Milli—Q (Phase A) water and 0.1% (*v v*^−1^) of formic acid in acetonitrile (Phase B), with a constant flow rate of 0.4 mL min^−1^. Detection by mass spectrometry was performed in positive ESI mode and negative ESI in full scan mode from 50 to 1100 *m z*^−1^ and MS/MS from 100 to 1700 *m*/*z*. Throughout the analysis, for mass correction the reference masses were used: *m z*^−1^ 121.0509 (C_5_H_4_N_4_), *m z*^−1^ 922.0098 (C_18_H_18_O_6_N_3_P_3_F_24_) for positive mode and 112.9856 [C_2_O_2_F_3_ (NH_4_)] and *m z*^−1^ 1033.9881 (C_18_H_18_O_6_N_3_P_3_F_24_) for negative mode. The QTOF instrument was operated in 4 GHz (high resolution). The chromatographic elution gradient began with 30% B for up to 15 min and was increased to 98% in 2 min, remaining in these conditions until minute 19. After this time, it returned to the initial conditions in 1 min, lasting 5 min in equilibration.

### 4.3. Metabolomic Profile by GC-QTOF-MS

Freeze-dried samples were used for metabolomic characterization. A total of 5 mg of each sample was mixed with 200 µL of MeOH. The samples were mixed by vortex for 15 min, followed by 10 min in ultrasound. Finally, the samples were centrifuged at 1500 rpm for 10 min. The supernatant was filtered with PTFE membranes of 0.22 µm diameter.

Of the previously prepared extracts, 30 µL were dried in a SpeedVac for 1 h at 35 °C. Next, 20 µL of O-methoxyamine (Sigma-Aldrich, St. Louis, MO, USA) in pyridine (15 mg mL^−1^) (Sygma-Aldrich) was added, mixed in vortex at 3200 rpm for 5 min. The mixture was incubated in the dark at room temperature for 16 h. The silylation process was carried out by adding 20 µL of BSTFA—Bis(trimethylsilyl) trifluoreacetamide with 1% TMCS-trimethylchlorosilane (Sigma-Aldrich); it was then mixed in vortex for 5 min and incubated at 70 °C for 1 h. Finally, samples were allowed to cool at room temperature and 100 µL of methyl stearate in heptane (Thermo Fischer Scientific Chemicals, Waltham, MA, USA) was added as an internal standard (5 mg mL^−1^) and vortexed for 5 min at 3200 rpm. For data acquisition, an Agilent Technologies 7890B gas chromatograph was used, coupled with an Agilent Technologies GC/Q-TOF 7250 time of flight mass analyzer (Agilent), which was equipped with a split/splitless injection port (250 °C, ratio split 30) and an Agilent Technologies 7693A autosampler. The electron ionization (EI) source was operated at 70 eV. An Agilent Technologies J&W HP-5MS column (30 m, 0.25 mm, 0.25 µm) was used; the carrier gas was helium at a constant flow of 0.7 mL min^−1^. The oven temperature was programmed from 60 °C (1 min) 10 °C min^−1^ to 325 °C (10 min). The temperatures of the transfer line to the detector, the source filament, and the quadrupole were maintained at 280 °C, 230 °C, and 150 °C, respectively. Detection by mass spectrometry was carried out between 50 and 600 *m z*^−1^ at a speed of 5 spectra s^−1^.

### 4.4. Data Treatment and Metabolite Annotation

LC-MS data processing was performed with the Agilent Mass Hunter Profinder 10.0 software, using the recursive molecular extraction algorithm. The annotation of major molecular metabolites was carried out using different tools, such as exact mass, molecular formula, mass spectra, and retention times. Each metabolite is reported with the confidence level ID, according to the guidelines established by the Metabolomics Standards Initiative (MSI). To search for metabolites in the different databases of the CEU MASS MEDIATOR, a search was carried out in batch. To confirm the annotation of metabolites, the MS-DIAL software Version MS-DIAL 4.90 was used, as well as the following libraries from the MONA export MassBank of North America: MoNA—export-LC-MS-MS_Positive_Mode, MoNA—export-MassBank, and MoNA—export-Vaniya-Fiehn_Natural_Products library. Similarly, MS/MS analysis was performed using the Agilent Mass Hunter Qualitative software version B.10.00. Analysis was conducted using the auto MS/MS algorithm, performing a manual comparison between the spectra of experimental masses and those reported in silico.

GC-MS data processing was performed using the Agilent Mass Hunter Unknown Analysis B.10.00 and the identification was performed by searching two specific libraries, namely Fiehn GC-MS Metabolomics RTL Library version 2013 and NIST Mass Spectral Reference Library (National Institute of Standards and Technology) version 2017. Metabolites were identified based on matching mass spectra and retention times with reference standards and library entries and by retention index (RI) calculations using homologous series of C7-C20 n-alkanes and fatty acid methyl esters (FAMEs).

## 5. Conclusions

The comprehensive untargeted metabolomic profile of *E. oleracea*, *E. precatoria*, *M. flexuosa*, and *T. grandiflorum* fruits from Mitú, Vaupés, Colombia, revealed distinct chemical signatures shaped by species and maturation stage. A key finding of this study was the high relative abundance of galacturonic acid in *M. flexuosa* pulp from Vaupés. Notably, our study also detected xylose and maltose at appreciable levels in *M. flexuosa*, which have been scarcely reported in regional literature, suggesting the potential for underexplored sugar diversity in Colombian populations. In the case of *T. grandiflorum*, galactose and citric acid were confirmed as predominant carbohydrates and organic acids, respectively, in both pulp and seed tissues. Flavonoid profile revealed a clear dominance of fisetin and kaempferol-O-rutinoside in ripe *E. oleracea* and *E. precatoria* fruits, with concentrations of significantly higher value than those reported in most Brazilian studies, where this flavonol is rarely identified. These findings suggest that Colombian açai may possess a distinct phenolic fingerprint, with potential applications for its antioxidant and therapeutic properties. The consistent presence of cyanidin-O-galactoside and isovitexin in both *Euterpe* species further supports their pharmacological relevance. Additionally, the detection of high concentrations of quinic acid and mannitol in the unripe stages of *Euterpe* and *Mauritia* species expands our understanding of polyol accumulation dynamics during maturation. The identification of myo-inositol and shikimic acid in both *T. grandiflorum* and *M. flexuosa*, along with regional variability in their concentrations, underscores the importance of conducting localized metabolomic surveys. By linking food composition to nutritional endpoints, this work supports the valorization of Amazonian biodiversity as a source of bioactive food with potential applications in functional food development, nutraceuticals, and dietary interventions.

## Figures and Tables

**Figure 1 ijms-27-00410-f001:**
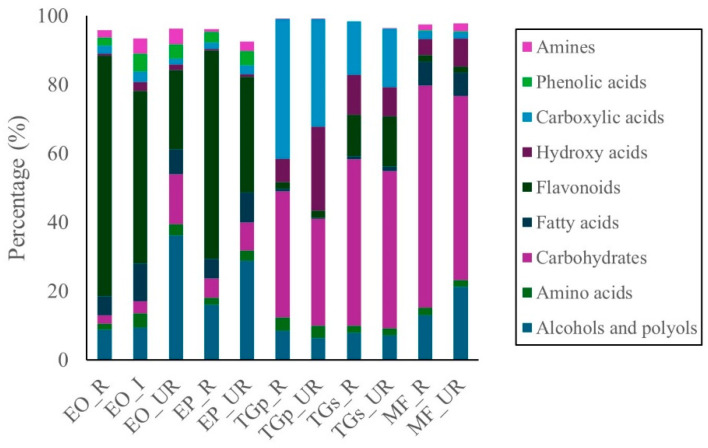
Metabolomic profiles of different Amazonian fruits: açai (EO_R: *E. oleracea* ripe, EO_I: *E. oleracea* intermediate, EO_UR: *E. oleracea* unripe), (EP_R: *E. precatoria* ripe, EP_UR: *E. precatoria* unripe), cupuassu (TGp_R: *T. grandiflorum* pulp ripe, TGp_UR: *T. grandiflorum* pulp unripe, TGs_R: *T. grandiflorum* seeds ripe, TGs_UR: *T. grandiflorum* seeds unripe), and mirití (MF_R: *M. flexuosa* ripe, MF_UR: *M. flexuosa* unripe).

**Figure 2 ijms-27-00410-f002:**
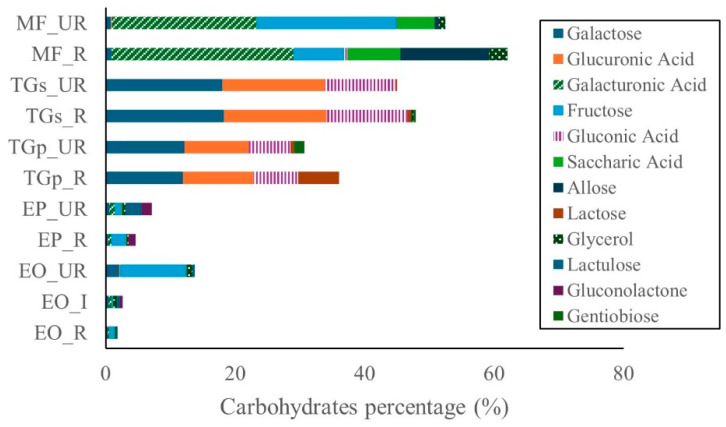
Carbohydrates in Amazon fruits: açai (EO_R: *E. oleracea* ripe, EO_I: *E. oleracea* intermediate, EO_UR: *E. oleracea* unripe), (EP_R: *E. precatoria* ripe, EP_UR: *E. precatoria* unripe), cupuassu (TGp_R: *T. grandiflorum* pulp ripe, TGp_UR: *T. grandiflorum* pulp unripe, TGs_R: *T. grandiflorum* seeds ripe, TGs_UR: *T. grandiflorum* seeds unripe), and mirití (MF_R: *M. flexuosa* ripe, MF_UR: *M. flexuosa* unripe).

**Figure 3 ijms-27-00410-f003:**
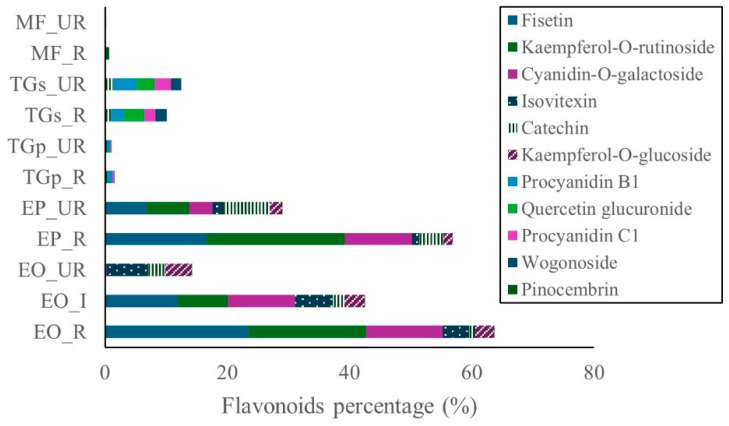
Flavonoids in Amazon fruits: açai (EO_R: *E. oleracea* ripe, EO_I: *E. oleracea* intermediate, EO_UR: *E. oleracea* unripe), (EP_R: *E. precatoria* ripe, EP_UR: *E. precatoria* unripe), cupuassu (TGp_R: *T. grandiflorum* pulp ripe, TGp_UR: *T. grandiflorum* pulp unripe, TGs_R: *T. grandiflorum* seeds ripe, TGs_UR: *T. grandiflorum* seeds unripe), and mirití (MF_R: *M. flexuosa* ripe, MF_UR: *M. flexuosa* unripe).

**Figure 4 ijms-27-00410-f004:**
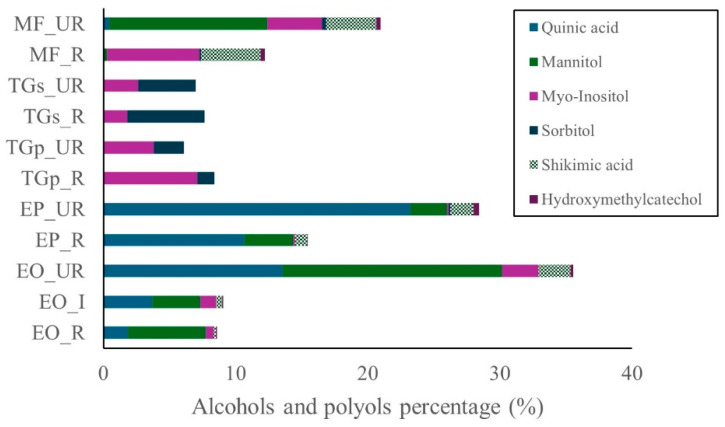
Alcohols and polyols in Amazon fruits: açai (EO_R: *E. oleracea* ripe, EO_I: *E. oleracea* intermediate, EO_UR: *E. oleracea* unripe), (EP_R: *E. precatoria* ripe, EP_UR: *E. precatoria* unripe), cupuassu (TGp_R: *T. grandiflorum* pulp ripe, TGp_UR: *T. grandiflorum* pulp unripe, TGs_R: *T. grandiflorum* seeds ripe, TGs_UR: *T. grandiflorum* seeds unripe), and mirití (MF_R: *M. flexuosa* ripe, MF_UR: *M. flexuosa* unripe.

**Figure 5 ijms-27-00410-f005:**
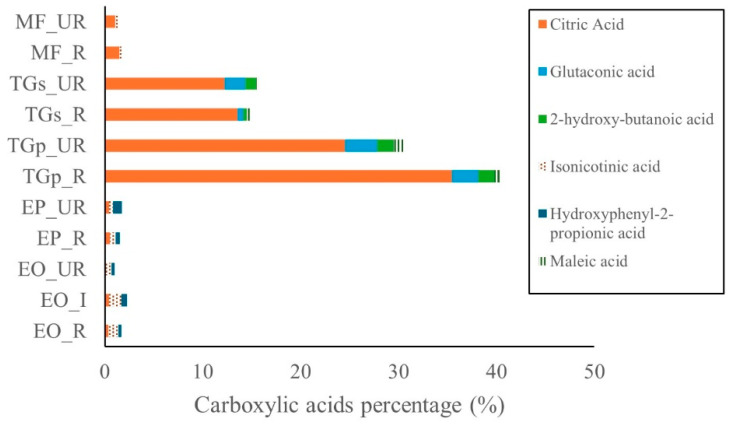
Carboxyllic acids in Amazon fruits: açai (EO_R: *E. oleracea* ripe, EO_I: *E. oleracea* intermediate, EO_UR: *E. oleracea* unripe), (EP_R: *E. precatoria* ripe, EP_UR: *E. precatoria* unripe), cupuassu (TGp_R: *T. grandiflorum* pulp ripe, TGp_UR: *T. grandiflorum* pulp unripe, TGs_R: *T. grandiflorum* seeds ripe, TGs_UR: *T. grandiflorum* seeds unripe), and mirití (MF_R: *M. flexuosa* ripe, MF_UR: *M. flexuosa* unripe.

**Table 1 ijms-27-00410-t001:** Fruit samples and ripening stages.

Species	Fruit Part	Ripening Stage	Fruit Color	Photographs	Sample Code
*E. oleracea* (açai)	Pulp	Ripe	Deep purple	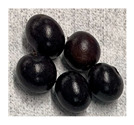 *Ripe açai.*	EO_R
		Intermediate	Light purple with green tones	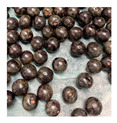 *Intermediate açai*	EO_I
		Unripe	Green	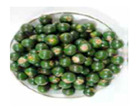 *Unripe açai*	EO_UR
*E. precatoria* (açai)	Pulp	Ripe	Deep purple		EP_R
		Unripe	Green		EP_UR
*M. flexuosa* (miriti)	Pulp	Ripe	Orange	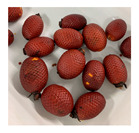 *Ripe miriti*	MF_R
		Unripe	Green scales in more than 50% of the fruit	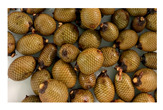 *Unripe miriti*	MF_UR
*T. grandiflorum* (cupuassu)	Pulp	Ripe	Brown	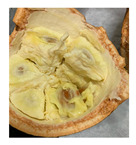 *Ripe pulp cupuassu*	TGp_R
		Unripe		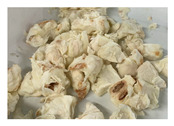 *Unripe pulp cupuassu*	TGp_UR
	seeds	Ripe		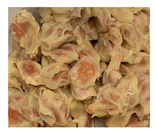 *Ripe seeds cupuassu*	TGs_R
		Unripe		No notable difference was detected	TGs_UR
Total samples			11		

EO_R: *E. oleraceae* ripe; EO_I: *E. oleracea* intermediate; EO:UR: *E. oleraceae* unripe; EP_R: *E. precatoria* ripe; EP_UR: *E. precatoria* unripe; MF_R: *M. flexuosa* ripe; MF_UR: *M. flexuosa* unripe; TGp_R: *T. grandiflorum* pulp ripe; TGP_UR: *T. grandiflorum* pulp unripe; TGs_R: *T. grandiflorum* seeds ripe; TGs_UR: *T. grandiflorum* seeds unripe.

## Data Availability

The original contributions presented in this study are included in the article/Appendix A. Further inquiries can be directed to the corresponding author.

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
