# Peer review of "Metabolomic Profile of Açai (Euterpe oleracea Mart., Euterpe precatoria Mart.), Mirití (Mauritia flexuosa L.), and Cupuassu (Theobroma grandiflorum (Wild. ex Spreng.) Schum) from Colombian Amazon: Insights into Nutritional Composition and Ripening Dynamics"

_ijms, 2025, doi:10.3390/ijms27010410_

Round 1
Reviewer 1 Report
Comments and Suggestions for Authors
The authors of the manuscript ‘Metabolomic profiling of açai (Euterpe oleracea Mart., Euterpe precatoria Mart.), mirití (Mauritia flexuosa L.), and cupuassu (Theobroma grandiflorum (Wild. ex Spreng.) Schum) from Colombian Amazon: Insights into nutritional composition and ripening dynamics’ analyzed untargeted bioactive compounds in three Amazonian fruits using LC-QTOF-MS and GC-QTOF-MS. The results are good; however, the authors failed to distinguish the compounds identified from the fruits and seeds using LC-QTOF-MS and GC-QTOF-MS.
Introduction: Please describe how ripening stages affect the bioactive contents. It would be helpful if the authors could explain how the ripening stage influences marketability and consumer preferences and indicate any challenges or limitations in studying bioactive compounds during fruit ripening.
L49-53: ‘Additionally, bioactive compounds with antioxidant, immunomodulatory, and anti-inflammatory properties.’ Citation needed.
L80-82: ‘Over the last decade, potential in food, pharmaceutical, and cosmetic applications.’ Citation needed.
Results: Please provide supporting LC-MS (positive and negative mode) and GC-MS chromatograms for all samples (Supplementary file). Also, please specify the compounds identified using LC-MS and GC-MS for better clarity.
L105: (Error! Reference source not found.). Figure 1.
L107: E. precatoria (Italic). Please italicize the species name throughout the text.
L163: Error! Reference source not found., Figure 4.
L167: (Error! Reference source not found.). Figure 4.
Discussion: Please indicate the role of the identified compounds in the ripening process.
L204: ‘11 to 49 g 100 g-1 in pulp’ Fresh weight or dry weight? Please mention it throughout the discussion.
L233: Kodikara et al. [17] highlighted
Materials and Methods: Please provide photographs of the samples (Table 1) for better understanding.
Conclusions: It is lengthy and could be better organized. Moving some discussions to the main section would allow for deeper exploration. This helps summarize key findings and implications effectively without detailed explanations better suited for the discussion.
L393: Please indicate the details of the supplementary tables.
Reviewer 2 Report
Comments and Suggestions for Authors
In this work, the authors studied the metabolomic profile of açaí, mirití, and cupuassu fruits from the Colombian Amazon. The paper is interesting; however, in my opinion, it requires substantial revision on several points:
- As the work focuses on fruit composition, the appropriate term would be metabolomic profile rather than metabolomic profiling, which appears in the title of the article and throughout the manuscript;
- It would be interesting to include, in the Introduction, information on the seed composition of these fruits, as this part has already been studied in previous works;
- Why did the authors not investigate the fruit peels?
- There are issues with the references in lines 105, 163, and 167;
- Please correct the unit formatting to g⁻¹ instead of g-1 in line 203 and throughout the manuscript;
- The paragraph that should start at line 223 appears to be missing. Please revise;
- In line 280, if possible, please add the technical specifications of the pulp extraction machine;
- Line 292 appears to be missing and should be revised;
- Table S1: What information is contained in the last two columns? What are the units?
- How was the quantification of these compounds carried out?
- It would be important to include, at least in the supplementary material, the main fragment ions used for compound identification in all fruits studied.
Round 2
Reviewer 1 Report
Comments and Suggestions for Authors
The authors have successfully addressed my comments and improved the manuscript.
Author Response
Thank you for your comments.
Reviewer 2 Report
Comments and Suggestions for Authors
The authors answered my questions satisfactorily. There is only a minor mistake in the legend of Tables S1, S2, and S3. Please correct “AR” to “RA” and add the meaning of the acronym.
Author Response
Thank you the mistake was corrected in the tables S1,S2 and S3